# Sustainable Nutrition for Increased Food Security Related to Romanian Consumers’ Behavior

**DOI:** 10.3390/nu14224892

**Published:** 2022-11-19

**Authors:** Ioana Mihaela Balan, Emanuela Diana Gherman, Remus Gherman, Ioan Brad, Raul Pascalau, Gabriela Popescu, Teodor Ioan Trasca

**Affiliations:** 1Faculty of Management and Rural Tourism, University of Life Sciences “King Michael I”, 300645 Timisoara, Romania; 2Faculty of Agriculture, University of Life Sciences “King Michael I”, 300645 Timisoara, Romania; 3Faculty of Food Engineering, University of Life Sciences “King Michael I”, 300645 Timisoara, Romania

**Keywords:** food security, food waste, overconsumption, sustainable and healthy food choices/systems, sustainable nutrition

## Abstract

Food security is a matter of global concern, as the supply of food is one of the basic needs, ensuring the survival of the species. The trend of globalization and development of the global economy has shifted the responsible, local consumption patterns towards an increased homogeneity of diets, with food products being disconnected from their source, leading to two major results: (1) increased global consumption and (2) increased uncertainty in the supply chain. To determine what is the nutritional model of Romanians, we developed a questionnaire of 32 questions and distributed it using simple random sampling method. The questionnaire was issued both in physical and digital form and received 1053 responses. The survey was conducted during 2021 and 2022, both in urban and rural areas with the aim of investigating the consumption pattern of the population. The analysis of the questionnaire data reveals overconsumption of animal products, starchy vegetables and bread and pastry products. This nutritional pattern with a high intake in animal protein, correlated with a lack of diversification, is extremely unsustainable, having a negative impact on human health and environmental health.

## 1. Introduction

In recent decades, the global agricultural production has grown at a faster rate than the human population, and today, available food, in terms of macronutrients, exceeds the global requirements, providing the opportunity to feed even more people [1,2]. The current global population growth rate is 1.05% per year and falling, while farmers today produce 262% more than in 1950 [3,4]. It is ironic, given the circumstances, that in 2020 three billion people are malnourished, of which between 720 and 811 million are undernourished [5], and the prevalence of undernutrition is on an increasing trend since 2019, from 8.0% in 2019 at 9.8% in 2021 [6]. Food security remains a global concern, despite all efforts and resources to end it, mainly because the level of food security is not only determined by the agricultural production. Food security represents a sum of interconnected factors, called in the specialized literature as the four pillars of food security. Each pillar makes a unique contribution to security, and any disruptions or issues in one, ripple through all the others. The four pillars of food security are:*Availability*—related to food supply, dependent on production, imports, storage;*Access*—both physical and financial access to available food products;*Utilization*—quality and safety of food resources, nutritional value, preparation;

and


*Stability*—the constant availability, access, and use of food resources over time [7].


In the self-sufficient historical communities, food resources were provided locally, based on available environmental resources: land fertility, water sources, duration of exposure to sunlight. Food resources were provided from agricultural practices as well as foraging, hunting, and fishing. The agricultural systems of the historical communities have been closely linked to their territory, one of the main concerns being to ensure the viability of primary resources for many generations [8,9].

The knowledge system and the sustainable management of resources have ensured long-term practices aimed at environmental conservation, highlighting the importance of adapting agricultural systems to different environments. Such an agricultural micro-cosmos reflects people’s ability and ingenuity to adapt to changes in their habitat, leaving lasting evidence of their commitment to the conservation of the agricultural heritage. These management systems, combining ingenuity and time-tested practices, have ensured food security in the community, along with the conservation of biodiversity and natural resources [10,11,12,13].

The industrial revolution drastically changed the agricultural practices, introducing machinery, chemical inputs, and standardized practices in rural areas, all to the benefit of increased production. With the development of the industrial revolution, many communities lost their agricultural heritage, the orientation towards increased production and uniformity leading to a loss of biodiversity and a change in perspective, from adapting production to the environment to adapting the environment to production [13]. Nowadays, agricultural systems focus primarily on staple food production, with four crops providing more than 60% of the globe’s energy intake: wheat, rice, maize, and potatoes. Increased uniformity, ease of transportation and storage translates into less diversified production and more uniform diets that lack nutritional balance and diverse nutrient intake [14].

In Romania, such an orientation towards production growth was intensely manifested during the communist regime, when agriculture became one of the main engines of economic growth, along with the metallurgical industry. The major problem with agricultural practices in Romania at the time was that, while Romania had been considered the “granary of Europe” since the interwar period and one of Europe’s all-time leading suppliers of grain, its citizens suffered from food insecurity, nutrient deficiency, and many related health conditions. Thirty years later, the disruption of the communist-era food supply still influences the population’s dietary patterns [15]. The “traditional Romanian diet” is largely based on meat, especially pork, bread and potatoes, a traditional tree-course menu exceeding 845 kJ (about 200 kcal) per 100 g per person. Given that the weight of such a menu is around 1000 g, based on the menus available online from Romanian restaurants, this means consuming at least 8450 kJ, or 2000 kcal, in one sitting. This trend of overconsumption among Romanian population represents the main point of interest for the survey carried out by the authors, and the results are in line with previous observational data regarding overconsumption of high caloric foods—meat, dairy, starchy vegetables and bread and pastry products. Not surprisingly, given the circumstances, that 51% of the country’s adult population is overweight and 19.1% is obese [16]. Furthermore, given the prevalence of obesity and overweight in adult population, a diet that relies heavily on meat consumption is extremely unsustainable [1,17]. 

Sustainable food consumption patterns have beneficial effects not only on human health but can also positively influence the impact of human population on the environment. A global disruption of the current consumption pattern, leading to a healthier diet and the consumption of culturally appropriate and nutritious local, seasonal foods, at every stage of human development, has the potential to reverse the current global health crisis related to overweight and obesity, and to restore the natural environment, without endangering the security of food resources, for current and future generations [17,18].

Another aspect worth mentioning in terms of sustainable production and consumption is the fact that currently, the disruption of climate patterns due to intensive agricultural practices will create further disruptions in the agricultural sector. Although only speculated at this point, the negative effects of climate change are already visible: an estimated 15 billion tons of fertile soil is lost annually do to land degradation and the incidence of extreme floods has increased by 50% in the last 10 years. The current trend of climate change, together with a growing urban population, translates into increased pressure on the rural environment, to support the growing, urban population [19].

In the same category as overproduction and overconsumption, threatening global food security, is food waste—an aggravating issue that has received increased attention from both the academic and political institutions. Even through food loss and waste (FLW) is not clearly defined by legislation, the amount of FLW is estimated to be roughly one third of the global food production, or about 1.3 billion tones annually [20]. Although FLW is recognized as an environmental issue, the nutritional value of lost and wasted food is rarely discussed [21]. Although numerous possibilities to prevent or reduce food waste at household level exist (correct planning of meals and portions, shopping list, correct storage of food, consuming leftovers as opposed to throwing), they are not quantified nor endorsed, being merely recommendations. To achieve global food security for present and future generations, measures are needed not only to increase the sustainability of production and consumption, but also to reduce loss and waste along the entire food chain.

The present research aims to identify if the consumption pattern of Romanians is a sustainable one, or if it aligns with the current habit of overconsumption, especially of food of animal origin.

## 2. Materials and Methods

The research bases both on secondary data, derived from consulting existing literature, and primary data collected by the authors. 

We created a questionnaire using the tool Google forms, owned by Alphabet Inc., Mountain View, CA, USA [22], as an instrument used to collect primary data, focusing the main food groups proposed by the EAT-Lancet Commission’s report, Food in the Anthropocene [17].

In order to have a valid measurement tool, we first identified the values that are relevant to our study: the frequency of food consumption during a week, per food groups and foods: daily, 4–6 times, 2–3 times, once, never;

and


the level of consumption/food group/person, during a week, measured in grams (g) and it’s multiple—kilogram (kg) or milliliters (mL) and it’s multiple—liter (L) where it was necessary.


The food groups and the foods included in the respective food group were determined by following the classification of the EAT-Lancet Commission [17] and those used by the Food and Agriculture Organization of the United Nations FAO in measuring dietary diversity [18]. The foods presented in the questionnaire have been adapted to correspond to real availability identified in Romania and grouped to compress the questionnaire to an acceptable length:from the “Tubers and starchy vegetables” [17,18] we eliminated cassava, as it is not available on Romanian market;vegetable consumption was measured for all vegetables together, as opposed to their grouping in “dark green”, “red and orange” and “other” [17] or “vitamin rich vegetables and tubers”, dark green leafy vegetables” and “other” [18];protein sources were split into two different categories: “animal products” and “legumes” [17]. Furthermore, tree nuts were included in a separate section: “nuts and seeds”;

and


for categories that cover a large pallet of foods, such as nuts and seeds, we introduced a conditional question: “Have you consumed other nuts and seeds in the previous week?”. If the respondent answered “yes” they would access a section where they were asked to mention what other nuts or seeds, they consumed and in what quantity.


In order to test the validity and reliability of the questionnaire, we submitted it to two different test rounds, conducted by two different groups of colleagues, both with and without academic background, before launching it publicly. We tested for clarity of expression, ease of completion, logical sequence of sections and questions, potential ambiguities regarding the time span covered by the queries (1 week), potential missing questions important to the inquiry, violation of privacy, viability to be used for future assessments and any other issues identified by the testing groups.

The method used for sampling was simple random sampling, individuals being chosen randomly, each member of the population having equal chances of being selected as respondent.

Upon launching the questionnaire to the public, it was published on multiple webpages, where volunteers could fill it. Simultaneously, we promoted the questionnaire on social media and collected physical responses from population. All respondents participated voluntarily, having the option to refuse or to renounce during completation.

The questionnaire has 32 questions divided into 8 sections: an informative section, six sections to identify the frequency and quantity of food consumption per food groups (whole grains, fruits, vegetables—with special subgroup for tubers or starchy vegetables and legumes, nuts and seeds, animal products—with subgroups dairy foods and eggs and meat, added fats, added sugars) and one section to collect personal data, the only data relevant for the study being age and level of physical activity [22]. A total of 1053 responses were collected in 2021 and 2022, from both rural and urban areas of Romania. This set of data is synthesized in Table 1.

The primary data needed to assess Romanians’ consumption habits were corroborated with secondary data, through extensive review of existing literature. The authors also consulted online available studies related to nutrition, recommended intake, overeating, Western dietary pattern, using multiple keyword combination searches, such as “food consumption + health”, “food + health problems”, “diet + health problem”, “diet + health”, “overeating”, “overconsumption”, “overconsumption + sustainable”, “sustainable diet”, “sustainable consumption”, “sustainable nutrition”, “nutritional guidelines”, “nutritional needs”, “adequate nutrition” and “nutritional requirements”. Following literature selection, studies regarding the correlation between nutrition, health, environment, and food security were selected as relevant literature for the present study. One of the key studies used for assessing the nutritional adequacy of respondents’ dietary patterns was the EAT-Lancet Commission Report from 2019, Food in the Anthropocene [17]. National food guidelines, both from Romania and from other countries, were also consulted to determine the availability of nutritional information and the population’s access level of nutritional information [23,24,25,26].

## 3. Results 

Like any other population on Earth, human population is heavily dependent on natural resources. Conscious consumption of these resources is an important determinant of future food security, as we must learn to efficiently use scarce resources to fulfill current demands and consider the needs of future generations. Efficient use of resources requires adjusting production and consumption to the real needs, while minimizing loss and waste, to ensure sustainable food production, for current and future food security [27]. Not only are we not currently equipped to ensure sustainability for future generation, but we are far from ensuring food security in present, at a global level. Data shows that the number of people affected by food insecurity grew from 1.63 billion people in 2014 to 2 billion people in 2019. This is partly due to the inequitable distribution of resources mentioned above, with many people affected by food insecurity—1.9 billion, being in the Global South [19]. Furthermore, although ending hunger is the second of the 17 Sustainable Development Goals (SDGs), aiming to “End hunger, achieve food security and improved nutrition and promote sustainable agriculture”, the events unfolding in the last 3 years have caused numerous disruptions and setbacks, undermining the progress previously made [28]. From 2019 to 2020, the prevalence of undernutrition from 8.0% to 9.3% worldwide, being estimated that the number of people affected by hunger in 2021 has grown to about 828 million, 150 million more than in 2019 [6].

At the opposite end of hunger and undernutrition, in the developed North, food market globalization puts great pressure on the consumers to access available food products, displaying a rich assortment of food products, both fresh and processed, healthy and unhealthy. These constant assaults from the environment, along with the evolution of food science leading to an increased supply of highly processed foods, are determining a change in consumer behavior, leaning towards a pattern of overconsumption and unhealthy nutritional intake [29].

In terms of the equitable distribution of food products, the agri-food market plays an important role in ensuring sustainable patterns of production and consumption. The importance of the market in achieving the sustainability targets of the 2030 Agenda is demonstrated by their mention in three targets of the Sustainable Development Goal 2 (SDG2)—Target 2.3, Target 2.b and Target 2.c—aiming to ensure proper functioning of the agricultural market [30]. The increased production capacity held by the Global North can contribute to increased food security for all entire global population if an ideal functioning of the agri-food market can be achieved.

At the World Food Summit in 1996, the four dimensions that together contribute to a state of food security were emphasized: “all people, at all times have physical and economic access to sufficient safe and nutritious food, that meets their dietary needs and food preferences for an active and healthy life.”. After decades of fighting food insecurity and implementing measures to combat the threat factors, the global community is nowhere near eradicating food insecurity [31].

Romania is no exception in this respect. Thirty years after the fall of the communist regime and the liberalization of the market, 13.9% of the population lives in moderate or severe food insecurity, while 6.6% of children under 5 are malnourished [16]. Malnutrition in early childhood can affect numerous aspects of later development of the future adult: cognition, language, social interaction—leading to lower academic performance, negatively impacting the human capital and causing social and economic problems [32]. For the Romanian population, both rural and urban, unaffected by food insecurity, the dietary pattern of choice is far from being sustainable and healthy [16,33].

The modern world, including Romania, faces “the double burden of malnutrition,” which manifests itself as both malnutrition and overnutrition [34]. Malnutrition is defined not only as a lack of macro- and micronutrient consumption, but also an excess intake, exceeding the human body’s energy requirements. In the Western, developed world, overconsumption is a manifestation of malnutrition, leading to a variety of health problems linked to an unbalanced diet, such as coronary heart disease, cancer, diabetes, and overweight and obesity, all of which are symptoms of underlying health problems [16,17,24,25,26,29,34].

Regardless of the health problems caused by inadequate nutrition and the overall cost of treating said health condition, the Western diet is extremely unsustainable [16,17,24,25,26,29,34]. Overconsumption of meat and meat products, fatty and sugary foods and drinks, highly processed food products and low intake of fresh fruit and vegetable are the main characteristics of the modern diet. The post-war, post-industrial background, of over-production and overconsumption, has generated serious economic and environmental issues, leading to a recurring discussion about food system sustainability and a challenge for food producers, as well as consumers, regarding the current consumption patterns. Recent studies in the field of sustainable practices in the food system reveal that one of the biggest contributions to the transition to sustainable nutrition, coming from the consumers, could be the transition to a plant-based, low-meat diet [16,17,24,25,26,29,35].

Consumer attitudes towards meat consumption are oriented in two directions, in line with global trends: consumers who are resistant to the idea of reducing the amount of meat and meat products consumed and consumers who, aware of the health and environmental consequences of meat consumption, choose to reduce or even eliminate meat from their diet [36]. Consumers who have adopted an extreme attitude and eliminated all animal products (dairy, eggs, honey) from their diet, embracing a vegan lifestyle, fall into the second category.

Many Romanian consumers tend to fall into the first category, mainly for hedonistic and cultural reasons. Romanian cuisine is a testimony over centuries of countless influences, from different cultures, who interacted on the current territory of the country (Dacians, Celts, Greeks, Romans, Byzantines, Slavs, Ottomans, Germans, Austrians, Hungarians, French, etc.). When it comes to meat consumption, preferences have evolved over the centuries, both under external influences and in the pedo-climatic conditions of the territory. After the crystallization of the cuisine, meat and meat products remained basic ingredients, often accompanied by vegetable products. Meat is usually eaten daily, several times a day: processed meats (ham, sausages, salami) at breakfast; a meat-based soup or broth and a main course consisting of meat (pork or chicken, grilled, battered, and fried, stewed or a stew of meat and vegetables, beans and sausages, meat cabbage rolls with sour cream, meatballs with sauce, meat stuffed peppers with sauce) for lunch; dinner may consist of leftovers from lunch or any of the dishes listed above. The preference for meat consumption at all three main meals of the day is noticeable and is also a habit that strongly contradicts the principles of healthy eating. However, this pattern is not representative for the entire Romanian population, trends among younger consumers (20–35 years) and those with access to education, leaning towards healthier habits. Given the Orthodox tradition of fasting in the months preceding important religious holidays (Easter, Christmas, Saint Mary) and the large number of Orthodox people—81% of the population, it can be considered that on rare occasions during the year, Romanians’ diet is plant-based and aims at a more diversified integration of plant-based dishes. This beneficial habit does not continue, however, at the end of the fasting period imposed by religious tradition [37,38].

The survey carried out by the authors reveals that the observational data presented above, from third party studies, are aligned with the self-reported consumption pattern of the respondents. For each food group considered in the questionnaire (whole grains, fruits, vegetables—with special subgroup for tubers or starchy vegetables and legumes, nuts and seeds, animal products—with subgroups: dairy foods, eggs, meat, added fats, added sugars) respondents had to provide answers to two questions: (1) one related to the frequency of consumption during the week, and (2) the other related to the quantity consumed during respective week, for each food. The frequency range was “daily”, “4–6 times”, “2–3 times”, “once” and “never”. As for quantity, the range has been customized for each food group, according to the recommended intake proposed by The EAT Lancet Commission [17]. 

Data analysis revealed that many respondents register increased frequency and quantity of consumption of the following food groups: bread and pastry products (63.53% of respondents consume more than recommended), tubers and starchy vegetables, especially potatoes (50.5% of respondents consume more than recommended) and meat and meat products, especially pork (70.9% of the respondents consume above the recommended quantity). These findings confirm the consumption pattern previously identified in studies regarding consumption behaviors in Romanian population. 

Furthermore, some of the following food groups registered very low frequency and quantity consumed during the week: fresh fruits and vegetables (except tubers and starchy vegetables)—79.34% of respondents consume under the recommended quantity, nuts, and seeds—81.4% of respondents consume less than recommended and fish and seafood—77.2% of respondents consume below the recommended quantity. The frequency of consumption, for fruits and vegetables and for animal products are presented in Figure 1. and Figure 2., respectively. The consumption behavior of respondents, for each food group, is detailed in Table 2. 

Regarding consumption of fresh fruits and vegetables, it is important to correlate the data obtained through questionnaire with the time in which the survey was conducted. Considering that the questionnaire was distributed during 2021 and 2022, it is possible that the low level of consumption registered for fruits and vegetables is determined by the decreased availability and increased prices of these foods on the marked, in the cold season. Although Christmas is preceded in Orthodox practice by a fasting period, for the responses timed in the first weeks of December, there was no decrease in consumption of animal products.

The data regarding consumption, gathered trough the questionnaire, reveal that the diet adopted by many of the respondents lacks proper nutrient intake, resulting from low consumption of plant-based foods and high consumption of animal products. Furthermore, the consumption pattern resulting from the survey reveals reduced diversification, both in terms of fruits and vegetables as well as animal products. 

In the vegetable group, potatoes register the highest level of consumption, and are more likely to be consumed more frequently, even compared to other starchy vegetables, such as beans and pulses. 

Regarding animal products, pork and pork products are consumed more frequently and in larger quantities during the week than any other animal products. On the other hand, fish and seafood, an excellent source of nutrients, are rarely eaten, with 55.8% of the respondents reporting they did not eat any fish and seafood during the reporting week.

Related to the specificity of nutrient consumption of the Romanian population, the Romanian National Institute of Statistic (INS) presents variations of these indicators, in the last two reporting years, respectively 2019 and 2020 [39]. According to the data provided by INS Romania, the average caloric intake of Romanian population was 3.548 in 2019, reaching 3.555 in 2020, a breakdown of the structure being presented in Table 3.

For the Romanian population, the consumption trend remained practically unchanged in 2020 compared to 2019, as presented in Figure 3, food insecurity being a localized issue, mainly within disadvantaged communities, such as low income or single income households, either from rural or urban areas [40]. Intriguingly, rural communities are equally affected by the incidence of food insecurity, even if the main descriptive feature of a rural area is the practice of agriculture. The situation can be determined by the fact that Romanian rural areas are mainly inhabited by aging population, unable to maintain agricultural activities, that rely on a low income. This situation is characteristic to most of the rural communities, regardless of the geographical location within Romanian territory [41].

## 4. Discussion

The lack of diversification and the focus on energy dense foods revealed by this study is not only specific to the Romanian population, but to the entire developed world. The consumption pattern identified through the questionnaire, focused on high intake of animal products at the expense of vegetables and fruits is characteristic of the Western dietary pattern, and has been the subject of many health-oriented studies [42,43,44,45]. 

In order to understand how the current consumption pattern of the Western world affects both human health and environmental health, a quick review of the elements of the Western diet and their impact is necessary: characterized by energy-dense food products, overconsumption of meats and dairy products, foods with a high content of saturated fats and sugars, excessive snacking and a low share of micronutrients and fibre, the Western diet is known for its adverse effects on human health. Romanian population is no exception to the rule, the study carried out by the authors revealing the same consumption behavior. High cholesterol levels, leading to coronary heart disease, high acidity which can increase the risk of cancer, liver problems, kidney problems and heart failure, high sugar intake which causes high blood pressure, inflammation, weight gain, fatty liver disease and diabetes, all linked to an increased risk of heart attack or stroke are elements of a typical Western diet [17,46,47,48]. 

The above-mentioned characteristics, elements and impacts of this pattern not only affect human health, but also put great pressure on the environment, mainly due to the pollution and resource depletion resulting from animal farming [17,49]. The environmental impact of animal farming, expressed in carbon footprint, land use and water use per kilogram of food product is presented in Table 4.

Considered as a share of environmental impact of all anthropogenic activities, agriculture uses 50% of the total living area, of which animal farming uses 77%, providing only 18% of the global calorie intake and 37% of the global protein intake. The graphic representation of this data is delivered in the Figure 4 [50].

To break this cycle of overconsumption and unsustainable dietary habits, the consumer society must be informed and adopt the alternative: a sustainable, healthy diet. This diet is not inherently new to society, with many communities relying on such a diet for centuries, with responsible food and environment behavior being part of the cultural heritage of those communities [17].

A healthy diet provides protection against malnutrition, regardless of its manifestation (undernutrition or overnutrition) and against non-communicable diseases (NCDs) resulting from improper nutritional intake (World Health Organization, WHO, August 2018). To improve the nutritional quality of foods, many of the basic foods (cereal, milk, pastas etc.) have been adapted to cover a wider range of needs, such as the case for fortified foods, for both children and adults [51]. For an increased impact of healthy dietary behavior, it is necessary to start from the beginning of life, so the education of parents and children is necessary to improve the quality of nutrition. The basic principles of a healthy diet, adopted by many of the WHO members states, which have committed themselves to combating malnutrition in all its forms, are the following:energy input must remain within the limits of energy consumption;the intake of fats, sugars, salts and alcohol should be limited;the intake of micro- and macronutrients should be ensured from plant based foods;meat, dairy products and eggs should be consumed occasionally and in small quantities;there should be a variety of foods consumed to ensure an adequate intake of nutrients;fresh foods should be prioritized by reducing processed foods and ready-to-eat meals;

and


the intake of liquids should be based on non-sugary drinks (water, natural fruit juices, tea, etc.) [52].


For a better understanding of these guidelines, they were adapted into graphic forms. Such a graphical representation is the AID-Food Pyramid, designed by the German Consumer Information Agency (Figure 5):-color coding for quantity—green means a lot of food, yellow for food that should be eaten in moderation and red represents food that should be eaten sparingly;-number of servings for each food;-what foods to eat.

From the first definition of sustainable diet as “food choices that support life and health within natural system limits into the foreseeable future” (Gussow and Clancy, 1986), to the definitions given by the International Scientific Symposium on Biodiversity and Sustainable Diets United Against Hunger in 2010 (organisez by The Food and Agricultural Organisation of the United Nations (FAO) together with WHO and Biodiversity International), the key elements of a sustainable diet, are shown in Figure 6 (Denis Lairon, President of the Federation of European Nutrition Societies) [53].

In 2019, the EAT-Lancet Commission formed by 19 commissioners and 18 co-authors from 16 countries, driven by the lack of scientific objectives to contribute to the formation of a sustainable food system, established the principles for a sustainable, healthy reference diet, adaptable to individual characteristics, social context and eating habits. According to the report, dietary changes from current consumption patterns to diets in line with the reference diet have the potential to prevent 10.8–11.6 million deaths per year, drive by a global reduction in the consumption of unhealthy foods. At least 50% of this reduction should be in the consumption of red meat and sugar, while at least doubling the consumption of vegetables, legumes, nuts, and fruits [17].

Regarding the structure of a healthy and sustainable diet, as shown in Figure 7, in correlation with the guidelines provided by the Commission, we can conclude [17]:-one third of the total energy intake should be based on whole grains, followed by vegetable protein sources, with ¼ of the total energy intake;-foods of animal origin should not exceed 12% of the total daily caloric intake for an adult;-sweeteners should be avoided or not exceed 31 g of added sugars per day per capita;

and


-unsaturated oils are recommended, while dairy fats should be avoided.


By correlating the data provided by numerous authors and public bodies mentioned above, regarding the characteristics of a sustainable diet, and the available data on the composition of a healthy diet, it can be concluded that they have many features in common:(a).focus on plenty of plant-based foods;(b).small quantities of meat and meat products;(c).low amount of fats and oil, emphasis on vegetable oils;(d).reduced amount of processed foods, focus on whole foods and increased consumption of fresh fruits and vegetables;

and


(e).quantity optimization and focus on quality.


Although sustainable diets and healthy diets share certain characteristics, the concept of a sustainable diet is much broader than that of a healthy diet and incorporates the latter. A healthy diet is characterized by the importance of macro and micronutrients and the nutritional value of the food consumed, resulting in increased well-being of the consumer, while a sustainable diet considers the impact of human dietary habits, healthy or unhealthy, on the environment, not only on human health [17,54].

## 5. Conclusions

A sustainable diet has many benefits for both the environment and human health. As sustainable diets emerge from sustainable agricultural practices, many of the food products are healthier than conventional agricultural products due to, among other reasons, the lack of use of synthetic pest and herbicides control.

In addition to the “cleanliness” of food products, sustainable dietary practices require the rational use of resources, including food resources. This rational use translates into:-optimal quantity consumed and optimal combination for an increased nutritional intake correlated with the optimal caloric intake;-the potential for reuse of food products, for example leftovers;

and


-reducing food waste and using unavoidable waste in the production cycle—transformation of waste into compost, for example.


All the guidelines of a sustainable diet are in line with the recommendations for a healthy diet, and furthermore, adopting a sustainable diet has an impact not only on personal health but also on the environment.

However, dietary changes cannot be made instantly, as in many cases food intake also depends on availability and affordability of foodstuff. Even in the developed and developing economies, such as Romania, food security is not ensured for the entire population. 

The research conducted by the authors, regarding Romanians’ dietary patterns reveals:an unbalanced diet, favoring bread, potatoes and pork to the detriment of other foods;a reduced consumption of fresh fruits and vegetables;lean meats are eaten less often than pork;other sources of animal protein (eggs, dairy products) are consumed less often than pork;meat is consumed frequently during the week;vegetable protein (pulses, nuts and seeds) is rarely consumed, meat being preferred;

and


consumption of whole cereals is reduced.


Such a diet is unsustainable for both human health, and environmental health. The human body needs a wide range of macro- and micronutrients to function properly, and —as studies focused on dietary patterns, and especially the Western Dietary Pattern (WDP) have shown—this nutritional pattern is not able to provide the body with the necessary nutrients. Furthermore, such a pattern, associated with the WDP, is unsustainable for the environment, as it relies on aggressive agricultural practices, mono-cultures and intensive animal farming, practices that have been demonstrated to damage the environment.

## Figures and Tables

**Figure 1 nutrients-14-04892-f001:**
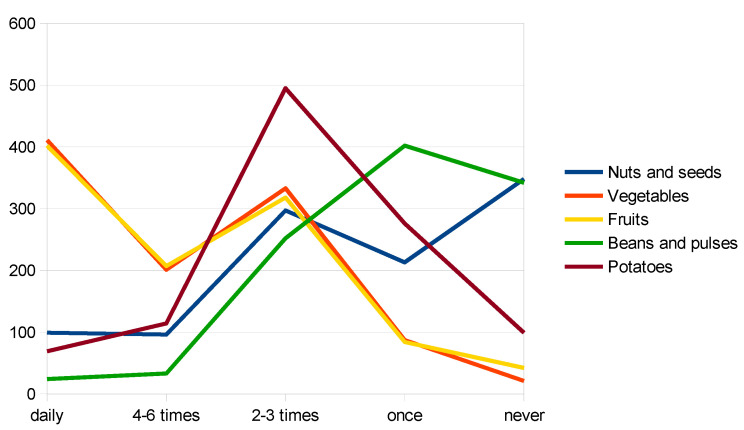
Frequency of fruits and vegetables consumption (authors’ study).

**Figure 2 nutrients-14-04892-f002:**
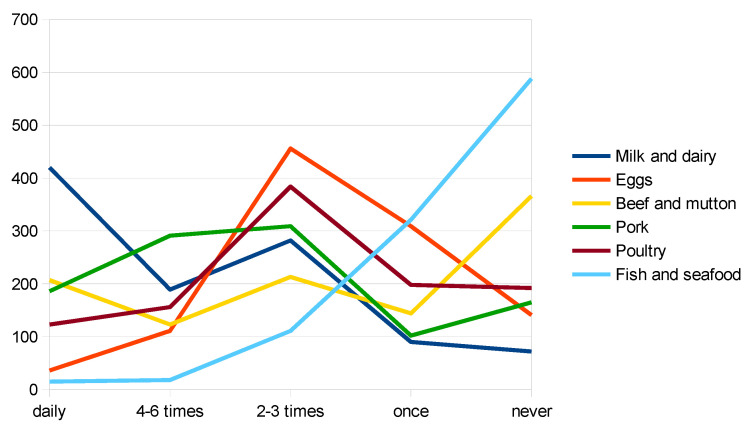
Frequency of animal products consumption (authors’ study).

**Figure 3 nutrients-14-04892-f003:**
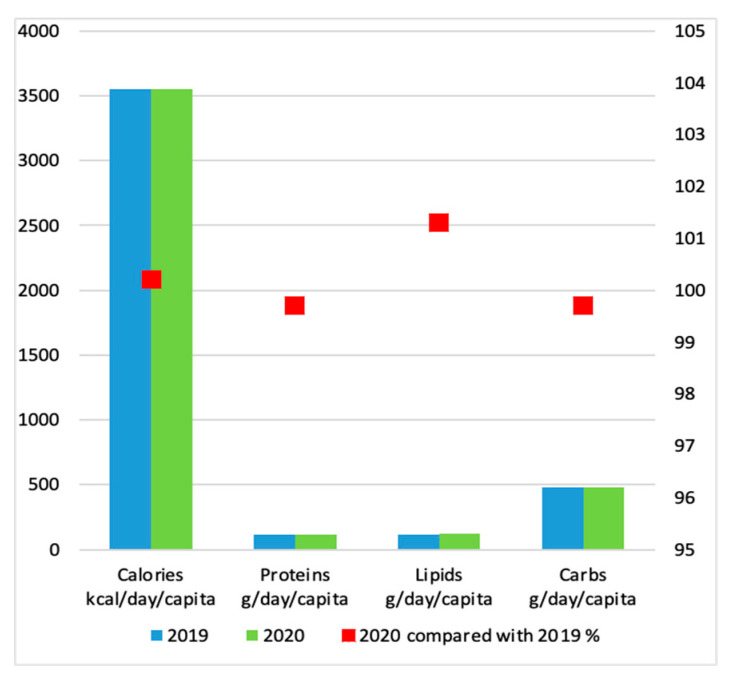
Total calories and total nutritional factors, in 2020 compared to 2019 in Romania of average daily food consumption per inhabitant [39].

**Figure 4 nutrients-14-04892-f004:**
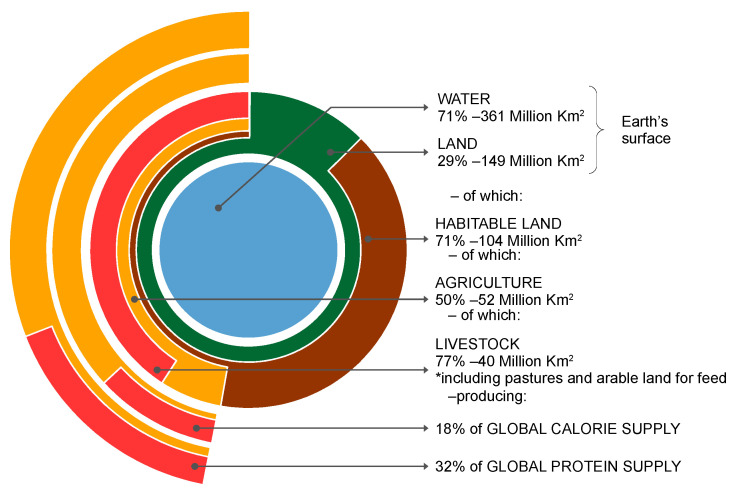
Land use per caloric outputs of animal farming as share of total—authors’ adaptation after [50]. * The land used for livestock includes farms, buildings and pastures and land for feed, not only the physical land occupied with animals.

**Figure 5 nutrients-14-04892-f005:**
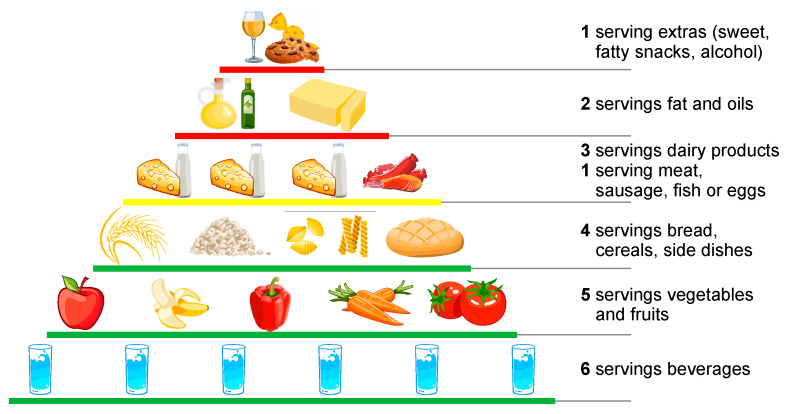
Recommended food intake (food pyramid)—authors’ adaption after [52].

**Figure 6 nutrients-14-04892-f006:**
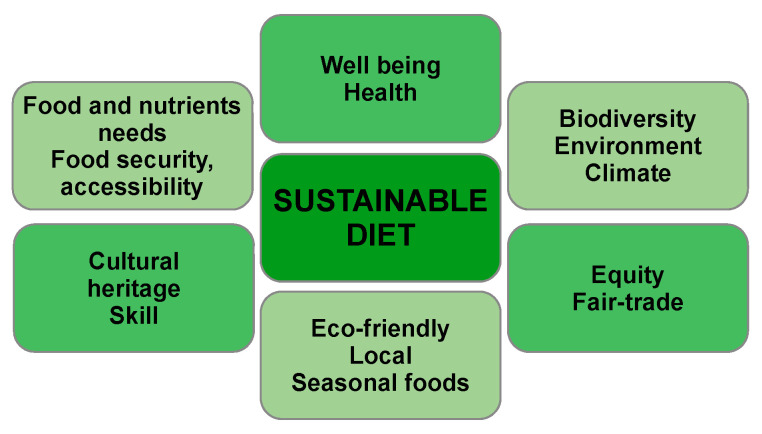
Key elements of a sustainable diet—authors’ adaptation of data from [53].

**Figure 7 nutrients-14-04892-f007:**
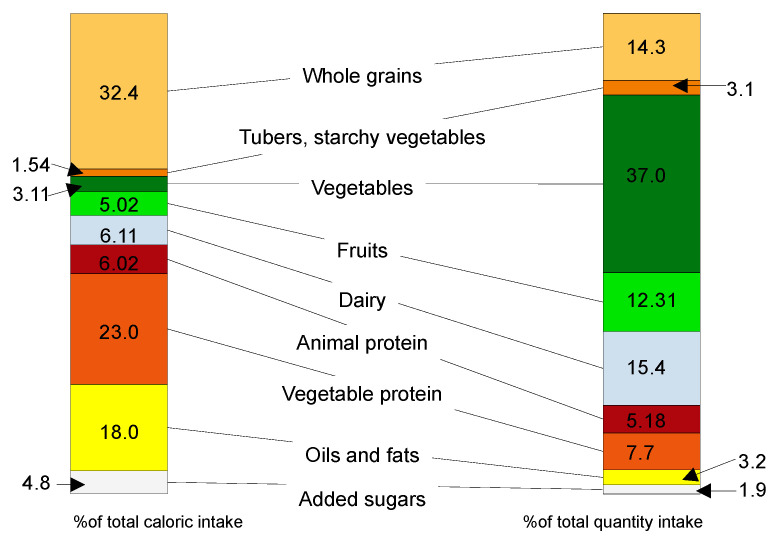
Caloric intake in relation to the amount of different food groups—authors own interpretation of data from [17].

**Table 1 nutrients-14-04892-t001:** Age and level of physical activity of the respondents.

Age Group	Number of Respondents	
	Absolute Value (−)	Relative Value (%)
15–20	117	11.1
21–30	576	54.7
31–40	156	14.8
41–50	135	12.8
51–60	63	6.0
>60	6	0.6
**Level of Physical Activity**		
*Sedentary—*no exercise (static job)	198	18.8
*Medium*–low intensity exercise (walking, house chores)	651	61.8
*Active–*moderate intensity exercise (exercising 2 to 3 times a week)	162	15.4
*Intense–*high intensity exercise (exercising 5 to 7 times a week)	42	4.0

**Table 2 nutrients-14-04892-t002:** Percent of respondents classified by consumption behavior, per food groups.

Foods and Food Groups	Under Consumption (%)	Normal Consumption (%)	Overconsumption (%)
Whole Grains	35.33	1.14	63.53
Fruits and Vegetables			
Potatoes	10.90	38.60	50.50
Beans and pulses	71.79	28.21	0.00
Fruits	67.80	29.40	2.80
Vegetables	73.30	7.10	19.60
Nuts and seeds	81.40	18.60	0.00
Animal products			
Milk and dairy	76.90	20.50	2.60
Eggs	26.50	50.10	23.40
Beef and mutton	69.50	12.70	17.80
Pork	15.40	13.70	70.90
Poultry	49.00	29.90	21.10
Fish and seafood	77.20	21.70	1.10
Added fats			
Vegetable fats	70.46	18.09	10.64
Animal fats	7.20	29.27	63.53
Added sugars	0.00	85.19	14.81

**Table 3 nutrients-14-04892-t003:** Average daily food consumption per inhabitant, expressed in calories and nutritional factors, in 2020 compared to 2019 in Romania [39].

Specification	Units	2019	2020	2020 Compared to 2019%
**Calories—total**	kcal/day/capita	**3548**	**3555**	**100.2**
-of animal origin	kcal/day/capita	970	971	100.1
-of vegetable origin	kcal/day/capita	2578	2584	100.2
**Proteins—total**	g/day/capita	**117.7**	**117.4**	**99.7**
-of animal origin	g/day/capita	62.3	62.3	100
-of vegetable origin	g/day/capita	55.4	55.1	99.5
**Lipids—total**	g/day/capita	**118.7**	**120.2**	**101.3**
-of animal origin	g/day/capita	62.1	62.2	100.2
-of vegetable origin	g/day/capita	56.6	58	102.5
**Carbs—total**	g/day/capita	**479.8**	**478.3**	**99.7**
-of animal origin	g/day/capita	34.5	34.5	100
-of vegetable origin	g/day/capita	445.3	443.8	99.7

The bold represent the total, the following numbers are the values (not bolded) that comprise the total.

**Table 4 nutrients-14-04892-t004:** Environmental impact of animal farming—authors’ selection after [50].

Environmental Impact of Food—Per Kilogram of Product	Beef (Beef Herd)	Beef (Dairy Herd)	Pig Meat	Poultry Meat	Lamb and Mutton	Prawns (Farmed)	Milk	Cheese	Eggs
Greenhouse gas emission (kgCO₂eq)	99.48	33.3	12.31	9.87	39.72	26.87	3.15	23.88	4.67
Land use (m^2^)	326.21	43.24	17.36	12.22	369.12	2.97	8.95	87.79	6.27
Freshwater use (L)	1451	2714	1796	660	1803	3515	628	5605	578

## Data Availability

Not applicable.

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
