# Peer review of "Sustainable Nutrition for Increased Food Security Related to Romanian Consumers’ Behavior"

_nutrients, 2022, doi:10.3390/nu14224892_

Round 1

Reviewer 1 Report (New Reviewer)

The manuscript is very interesting, well written and deals with a current issue. The introduction is well articulated and with adequate references. The methodology is also fairly illustrated. Despite these points in favor, however, the article is dispersive and too long in some places, distracting attention from the focus of the article and the results. Therefore, the part of the results and the discussion should be better organized; for example: in particular, the whole part of page 5 (line 174 to 219) should be shortened and contextualized in the introduction, because it detracts from the results. Similarly, the part on the habits of Romanians from line 250 to 281 should be summarized by adding some bibliographical references that are lacking in this part. always with regard to the discussion from lines 400 to 476 it has the same problem: too verbose and too much relevance to the results and to the discussion of the results and should be shortened.

Author Response

Dear Reviewer 1,

Thank you very much for the valuable appreciation on our manuscript and your comments and recommendations.

Please find in attachment our answers, point by point, to your comments.

Kind regards,

The authors

Reviewer 2 Report (New Reviewer)

The article is mainly a descriptive study on the consumption pattern of the Romanian population that its validity is questionable.

Abstract: The method is not fully explained

Introduction:

Considering the large number of the figures, figure 1 can be deleted.

The aim of the study has not been clarified at the end of introduction.

Methods:  The content of the questionnaire in the framework of Google form is neither created nor endorsed by Google (ref 54). Validity of the questionnaire in extracting dietary patterns are not clear. Without clarifying the validity of the data gathered, results are not reliable.

The questionnaire seems like the Dietary Diversity ones.

Some results are derived from existing data that have not mentioned in Methods. Results are not derived from the methods

Table 1 does not show the frequency of men and women and used value instead of frequencies.

Ethical considerations of the study is not clear.

Results and Discussion: Findings and discussion, instead of starting with the findings, started with several pages of explanations that can be shortened; e.g the following paragraph:

“Food security exists when all people, at all times, have physical and economic access 194 to sufficient safe and nutritious food that meets their dietary needs and food preferences 195 for an active and healthy life.” This widely accepted definition of food security was first 196 formulated at World Food Summit in 1996, emphasizing the four dimensions that together 197 contribute to a state of food security. After decades of fighting food insecurity and 198 implementing measures to combat the threat factors, the global community is nowhere 199 near eradicating food insecurity.

Using a line chart for figure 4 doesn't seem right.

Conclusion: is not based on the study results.

Author Response

Dear Reviewer 2,

Thank you for the comments, observations and recommendations which were very helpful for us to increase the quality of our article, as you will see below.

Please find attached our answers, point by point, to your comments.

Kind regards,

The authors

Reviewer 3 Report (New Reviewer)

The topic is of interest with an eye toward sustainability, but this study really focuses on following the nutritional guide. While that is of interest, it does wander from what the title suggests.

The "case study" approach is not really followed here. The structure in this work is an introduction, a short literature review, a short methodology section with very little treatment of the actual methods used to create an instrument, access respondents, etc., a results section that goes back to being a second literature review but then follows up with results. If there is methodology and an actual study, this should be a different type of manuscript rather than a "case study".

Extensive editing needed. The thought is always understood, but wording and grammar and punctuation need a great deal of work.

Author Response

Dear Reviewer 3,

Thank you for the appreciation on our manuscript and your comments. We took them into consideration to increase the quality of our article.

Please find in attachment our answers, point by point, to your comments.

Kind regards,

The authors

Reviewer 4 Report (New Reviewer)

In this paper, the authors summarize and analyze the food safety, sustainable diets, consumption patterns of the cases of Romania and put forward a series of recommended actions to change the current unsustainable consumer behaviors. I have the following comments:

1.     The article puts forwards a large number of nutritional discussion, but lacks relevant literature citations. Such as Line 229.

2.     Part 3 is too large compared to other chapters, results and discussion should be given separately.

3.     The “conclusions” part included the suggestion of reuse of food products, but it is not mentioned in the above text. It is suggested to add the part to the study.

4.     Table 4 should be re-written, the format of “environmental -per kilogram of product” is not proper.

5.     In the Material section, more information of statistical analysis should be added, and participants should be introduced. What kind of sampling method did the authors use? What are the reliability and validity of the questionnaire?

6.     English needs to improved.

7.     Figure 4 should be introduced in the text.

Author Response

Dear Reviewer 4,

Thank you for the valuable comments! We consider that helped us to introduce necessary changes in our article and to increase its quality.

Please find attached our answers, point by point, to your comments.

Kind regards,

The authors

Round 2

Reviewer 2 Report (New Reviewer)

Many improvements has been made.

Reviewer 3 Report (New Reviewer)

Thank you for the chance to review this revised version.

Reviewer 4 Report (New Reviewer)

All my questions are well addressed.

This manuscript is a resubmission of an earlier submission. The following is a list of the peer review reports and author responses from that submission.

Round 1

Reviewer 1 Report

1. The rubric "Material and methods" needs to be significantly expanded, in particular in the part of author's research.

2. The main remark: the article does not contain a specific author's empirical research.

3.Lines 294-295 state that "This section is not mandatory but can be added to the manuscript if the discussion is unusually long or complex." Why write this?

4. The reference must be checked and completed in accordance with the requirements.

Reviewer 2 Report

The manuscript entitled “Sustainable Consumption Pattern for Increased Food Security – Case Study Romania” touches an important research area. Authors have tried to ‘compile’ some data from ‘third-party’ materials. However, the overall quality of the manuscript is not up the mark. If it is to be considered for publication, major revisions are required. Please see my comments below:

The problem statement, research objectives, and research gap should be clearly stated at the end of Introduction section.

Material and Methods section is particularly weak. The details of data collection are obscured. The measurements and definitions should be operationalized, and the methodology should be clearly stated that what was the criterion for including the ‘third-party’ materials in the study. At the same time, the authors are expected to provide relevant citations.

Considering the above comments, the paper cannot be considered as a ‘research article’ since no proper methodology and data analysis have been employed. I would encourage the authors to rewrite it as a review paper and enhance the content. If it is to be considered as a research article, a solid methodology section should be included. For example, provide details for the methods used for conducting a qualitative study. See the following literature:

Li, T., et al. (2021). "Rethinking the Role of Grain Banks in China’s Agriculture." Agriculture 11(1): 49.

Furthermore, please add comprehensive literature review on all dimensions of food security, particularly in Romanian (as well as similar countries in Europe) context. When talking about sustainability, the authors also need to discuss the emerging foods which are either aimed to overcome micronutrient deficiencies (such as biofortified foods) or reduce the meat consumption (such as PBMAs). See the literature below:

Razzaq, A., et al. (2021). "Towards Sustainable Diets: Understanding the Cognitive Mechanism of Consumer Acceptance of Biofortified Foods and the Role of Nutrition Information." International Journal of Environmental Research and Public Health 18(3): 1175.

Harguess, Jamie M., Noe C. Crespo, and Mee Yong Hong. “Strategies to Reduce Meat Consumption: A Systematic Literature Review of Experimental Studies.” Appetite 144 (January 2020): 104478.

The discussion section is also missing from the article.

Overall, the authors have failed to write a convincing article. I would suggest a significant revision keeping in view the above comments.

Minor problems

Line 294-295: “This section is not mandatory but can be added to the manuscript if the discussion is unusually long or complex.”??

Line 71: Is it three-course menu?

Reviewer 3 Report

After having read carefully the article, I still cannot understand what it is adding to the existing literature. There is nothing original in the paper, all the drawings even are based on someone else's work and not really adapted to the Romanian context - it could be an essay (actually well written one!), but not a research article.